# Multistage Psychometric Testing of the Homeless Health Access to Care Tool

**DOI:** 10.3390/ijerph192315928

**Published:** 2022-11-29

**Authors:** Jane Currie, Elizabeth Grech, Jasmine Yee, Amy Aitkenhead, Lee Jones

**Affiliations:** 1School of Nursing, Queensland University of Technology, Brisbane, QLD 4000, Australia; 2Homeless Health Service, St Vincent’s Hospital Sydney, Sydney, NSW 2010, Australia; 3School of Public Health and Social Work, Queensland University of Technology, Brisbane, QLD 4000, Australia; 4QIMR Berghofer Medical Research Institute, Brisbane, QLD 4000, Australia

**Keywords:** access to care, homelessness, vulnerability, psychometric testing, validity, reliability, internal consistency

## Abstract

People experiencing homelessness find it challenging to access the healthcare they so desperately need. To address this, we have developed the Homeless Health Access to Care Tool, which assesses health related vulnerability (burden of injury and/or illness and ability to access healthcare) and can be used to prioritize people for access to healthcare. Here, we report the initial psychometrics of the Homeless Health Access to Care Tool. To assess interrater reliability, clinician participants were invited to instinctually rate the health-related vulnerability of 18-fictional case studies and then apply the Homeless Health Access to Care Tool to these same case studies. To assess convergent validity, the Homeless Health Access to Care Tool and the tool it was adapted from, the Australian Vulnerability Index Service Prioritization Decision Assistance Tool were administered to people experiencing homelessness. Feedback was sought from the participants receiving these two tools and from those administering them. The Homeless Health Access to Care Tool demonstrated a high interrater reliability and internal consistency. Participants using and receiving the Homeless Health Access to Care Tool reported it as straightforward, unintrusive and clear. Median time of administration was 7 min 29 s (SD 118.03 s). Convergent validity was established for the Homeless Health Access to Care Tool with a moderate correlation (r = 0.567) with the total score of the Australian Vulnerability Index Service Prioritization Decision Assistance Tool. Findings suggest that the Homeless Health Access to Care Tool is feasible and reliable. Larger samples are required to report construct validity.

## 1. Introduction

Evidence associating homelessness and morbidity and mortality is compelling. Being homeless or living in temporary and/or unstable accommodation has a negative impact on health outcomes and life expectancy [1]. Even one episode of any level of homelessness can result in premature mortality. A 15-year retrospective cohort study reported that homeless individuals had younger median age at death (66.60 vs. 78.19 years) and significantly increased mortality risk ratios compared to non-homeless individuals [1]. When experiencing homelessness people are subject to injury through accidents and assaults, as well as illnesses related to the conditions of rough sleeping such as respiratory infections, mental ill health, and skin infections and drug and alcohol use [2,3,4,5,6]. Although the health needs of people experiencing homelessness are often higher than those living in stable accommodation, this vulnerable population is less likely to seek healthcare, particularly primary healthcare [7]. When people experiencing homelessness do seek care, it is often at a later stage of ill health and more frequently via the emergency department, rather than through primary healthcare services. When admitted, people experiencing homelessness have longer, and therefore more costly, hospital admission and poorer health outcomes compared to those in stable accommodation [8]. The chronicity of many of their health conditions means people experiencing homelessness wait longer to be treated in an emergency department, they are more likely to leave without being seen and re-attend within 28 days [9,10,11]. A Melbourne study reported that 43% of people experiencing homelessness would re-attend the same emergency department within 28 days [10].

### 1.1. Background

We have developed a tool to assess health related vulnerability of people experiencing homelessness called the Homeless Health Access to Care Tool (HHACT). Health related vulnerability is defined as a person’s health need in terms of their burden of disease and injury, combined with their ability to access health care. Ability to access health care is defined using the definition proposed by Levesque et al. (2013) [12]: ability to seek, perceive, reach, pay and engage healthcare. Vulnerability indices are commonly used to assess mortality and morbidity of people experiencing homelessness [13]. Vulnerability indices populate a risk score based on the self-report responses to a series of questions on health and wellbeing. The score is used to triage and prioritize people for housing, but not for healthcare. Over the past two years, the HHACT has been developed by the Homeless Health Research Team at St Vincent’s Hospital Sydney. The HHACT is a 24-item questionnaire, adapted from an existing tool, the Australian Vulnerability Index Service Prioritization Decision Assistance Tool (VI-SPDAT) designed to prioritize people for housing. The HHACT results in a score, which classifies an individual as either low level of vulnerability, moderately vulnerable, or highly vulnerable. In future studies we intend to combine the HHACT with a decision assistance guide that will inform the onward care provided to people experiencing homelessness. By combining the HHACT with a decision assistance guide we intend that a comprehensive assessment is undertaken and that options for access to healthcare are maximised for each person experiencing or at risk of homelessness. The process by which the HHACT has been developed has been rigorous and dynamic, informed by the ‘Method for Developing and Validating Scales for Health, Social and Behavioral Research’ [14]. This Method comprises three phases that span nine steps. The first phase of development of the HHACT is reported elsewhere [15]. In this paper, Phase 2 Scale Development is reported, which includes pre-testing questions, survey administration and sample size, and item reduction.

### 1.2. Definition of Homeless

There is no universally accepted definition of homelessness. The Australian Bureau of Statistics’ [16] has defined homelessness to include the absence of suitable accommodation, living in an inadequate dwelling, having no or limited/non-extendable tenure, and having no control of/no access to or space for social relations. Homelessness is a broad term that is often used to describe the following population groups. (1) Rough sleeping; (2) Supported accommodation (e.g., refuges & crisis accommodation; (3) Short –term accommodation without tenure (e.g., boarding houses, hostel, caravan, couch surfing); and (4) Accommodation in an institutional setting (e.g., drug and alcohol rehabilitation centers, hospital, correctional facility) [17]. It is estimated that globally 100 million people are homeless and one in four people live in insufficient housing that impacts their health, safety, and prosperity [18]. The OECD Social Policy Division (2021) [19] reported 2.1 million people across 36 countries (from where data are available) as homeless. This is 567,715 people in the United States (US), and over 100,000 people experiencing homelessness each in Germany, France, Canada, Australia, and Brazil.

In Australia, the 2016 Census estimated 116,427 people were experiencing homelessness, Aboriginal and Torres Strait Islander peoples account for up to 20% of this population [16]. Establishing social distancing and isolation for people experiencing homelessness during the COVID-19 pandemic was challenging. Australian jurisdictions responded aggressively and housed up to 33,000 rough sleepers in hotels and other forms of temporary accommodation [20]. Now that pandemic measures have eased, and the economic impacts of the pandemic are being realized, a rise in homelessness is predicted [20]. In Australia, domestic and family violence is a leading cause of homelessness. The social isolation, social distancing and recurrent lockdowns resulted in an escalation in the incidence of domestic and family violence [21], which is also amplifying the presence of homelessness.

When attending an emergency department for what may appear to be routine healthcare, people experiencing homelessness are more likely to be assessed as a less urgent clinical priority than the general population, despite their complex healthcare needs [22]. There is debate as to the role that emergency departments can have in interrupting the cycle of homelessness. Proponents argue that because most people experiencing homelessness access emergency department care, their attendance is an opportunity not to be missed to link this vulnerable population with healthcare and other services [23,24]. Yet, evidence suggests that some emergency departments are under resourced to meet the myriad of health and social needs of people experiencing homelessness [25]. Despite the impact of homelessness on health outcomes, screening for homelessness is not mandated in Australian emergency departments and homelessness is often under recognized [10]. The absence of screening for homelessness has implications for appropriate discharge, likelihood of adherence with treatment [26], and is a missed opportunity for referral to support services. For people experiencing homelessness there are many reasons why they do not readily access primary and preventative healthcare. The competing demands of being homeless in relation to finding shelter and access to food can hamper any attempt they might make to attend primary healthcare or other types of appointment. The stigma they face when accessing healthcare can also be a substantial deterrent [27]. In developing the HHACT, we intend to address the challenges of healthcare access for people experiencing homelessness, by using the HHACT to prioritize access to healthcare, for use in emergency departments and community healthcare settings. Here, we report the initial psychometric testing of the HHACT.

Primary Objective: To assess the reliability and validity of the HHACT when applied to people experiencing homelessness. Secondary Objective: To explore the usability and feasibility of the HHACT.

## 2. Materials and Methods

An overview of study 1 and study 2 is shown in Figure 1. The design of each study is discussed further below.

### 2.1. Study 1: Interrater Reliability

A purposive cross-sectional sample of *n* = 44 clinicians practicing in the Homeless Health Service and Emergency Department at St Vincent’s Hospital Sydney were invited to instinctually evaluate the level of health-related vulnerability of 18 fictional case studies of people experiencing homelessness (Appendix A). Participants were purposefully selected based on their experience in providing healthcare to people experiencing homelessness. The case studies were designed by the lead author and independently reviewed by three clinicians, each with experience and expertise in homelessness: a registered nurse, nurse practitioner and emergency department specialist physician. In their instinctual evaluation, participants were asked to rate each case study as highly, moderately, or low level of health-related vulnerability. After each case study the participants were asked to provide any free-text comments in relation to the scoring of each case study. These free text comments provided a substantial amount of feedback regarding participants’ rationale underpinning their rating of each case study. Definitions of vulnerability were provided as follows:

*Highly vulnerable*—has multiple chronic, complex and or acute health needs and health is likely to deteriorate without urgent intervention. Will need one or more health care workers to support them to access healthcare (e.g., someone to transport them to appointments and stay with them throughout the consultation).

*Moderately vulnerable*—has several chronic, complex and or acute health needs and health might deteriorate without intervention. Will need verbal or written support to access healthcare and one on one support for some components of accessing healthcare (e.g., someone to stay with them during a healthcare consultation).

*Low level vulnerability*—has some chronic, complex and or acute health needs and health is unlikely to deteriorate without intervention. Will need some guidance in terms of verbal and written support to access healthcare (e.g., written or verbal information on how to make an appointment and can access the consultation on their own).

Once instinctually evaluated, the Homeless Health Research Team (*n* = 5), a group of clinicians, each with greater than three years of experience providing healthcare to people experiencing homelessness were invited by email to apply the HHACT to the same case studies and these results were compared to the instinctual ratings.

### 2.2. Procedure of Data Collection

Participants were approached by email, inviting them to participate in the instinctual review of the 18 case studies via a survey on the Survey monkey platform (Appendix A). Participants were provided with a Participant Information Sheet; consent was deemed through completion of the review of the case studies. The survey asked participants to rate each case study based on their perceived level of health-related vulnerability, which took approximately 30 min to complete. The participants were also asked to provide free text feedback regarding their scoring of each case study.

Once instinctually rated, members of the Homeless Health Research Team who are clinicians and have a high level of expertise in the field, were invited to apply the HHACT to each of the 18 case studies, via an online survey hosted by the Survey Monkey platform. These Homeless Health Research Team participants were purposively selected as they had an in depth understanding of the HHACT and consequently were best placed to apply it to the fictional case studies. Copies of the HAACT were provided in a Microsoft Excel spreadsheet and participants were asked to record their HHACT scores in the spreadsheet. It took between 30 to 45 min to apply the HHACT to the case studies. There was no financial compensation for participation in either survey, consent was voluntary or deemed by completion of the surveys. The lead author, an expert in homeless health, undertook an assessment of each case study both instinctually and then by applying the HHACT. This expert assessment was undertaken to enable a comparison of the performance of instinctual and HHACT assessments.

### 2.3. Procedure of Data Analysis

Survey data was input to SPSS version 27 (SPSS Inc., Chicago, IL, USA) for analysis. Two researchers independently collated the data into tables that compared the results of the instinctual surveys and the application of the HHACT to the case studies. The qualitative feedback relating to why a specific rating was chosen in the instinctual survey was analyzed using an adaptation of Braun and Clarke’s [28] thematic analysis process. In the first stage the first (JC) and second author (EG) became familiar with the data. The data was then coded by the second author and sorted into potential themes. These themes were then reviewed by the first and second author together and were then confirmed. Feedback directly relating to questions in the HHACT was presented to the Homeless Health Research Team over a series of three meetings during which the team discussed the feedback with a view to modifying the HHACT questions.

The quantitative analysis was undertaken by one author independently (LJ). Intraclass correlation (ICC) was used to explore the reliability of ratings for variables with a continuous score. Reliability was assessed using the ICC (2, 1), a two-way random model assuming absolute agreement with a single rater measurement. Good reliability can be seen as ≥0.7. The reliability of categorical variables was investigated using Gwets Kappa with linear weighting, this assesses agreement beyond chance and performs well in situations with high agreement.

### 2.4. Study 2 Feasibility, Convergent Validity

A Peer Support Worker (JY) with lived experience of homelessness and a nurse practitioner (AA), both members of the Homeless Health Research Team administered the HHACT (Table 1) and the Australian VI-SPDAT to *n* = 20 volunteers experiencing homelessness who were residing in either Tierney House or Stanford House, St Vincent’s Hospital Sydney. Tierney House is a step-down residential facility for people experiencing homelessness who have a primary-sub acute physical health issue/concern. Stanford House is also a residential facility that provides respite and support for people living with Human Immunodeficiency Virus (HIV) who are experiencing or at risk of homelessness.

Prior to recruiting the Tierney and Stanford House residents, a poster advertising the study and the dates of data collection were displayed. Eligible participants were those experiencing homelessness and were residing in Tierney or Stanford House for a minimum of 4 days prior to data collection and had the capacity to provide informed consent. Those interested in participating in the study had the opportunity to express their interest and ask questions about the study to staff members. On the days of data collection, the Peer Support Worker discussed the project with those who expressed interest and provided them with a copy of the Participant Information Sheet. All participants provided written consent to participate, and they were handed a $30 voucher on completion.

### 2.5. Procedure of Data Collection

The HHACT and the VI-SPAT were administered to *n* = 20 participants residing at Tierney or Stanford House by the same person (A.A.). The time taken to administer the HHACT was measured by the Peer Support Worker (J.Y.). Immediately after the HHACT was administered, the Peer Support Worker (J.Y.) administered a short feedback survey (Appendix A) to elicit participants first impressions of the HHACT. Once data collection had ceased, a semi-structured interview was conducted by the primary author (J.C.) with the colleagues administering the tools (A.A., J.Y.) to explore the feasibility of administering the HHACT. An email invitation was sent to each participant, attached was a participant information sheet. Participants provided written consent. The interview guide is shown in (Appendix A). The interview was recorded with permission from all participants. The qualitative data was professionally transcribed and de-identified for the purpose of data analysis and storage.

### 2.6. Procedure of Data Analysis

The scores from the completed HHACT, VI-SPDAT and feedback surveys were input to SPSS for analysis of convergent validity, which was assessed for the total score and subscales using spearman’s correlation, good convergent validity was seen as r ≥ 0.5. The participant HHACT feedback survey was input to SPSS and analyzed descriptively. The qualitative data collected from the focus group was analyzed using Braun and Clarke’s [9] thematic analysis approach as described above.

## 3. Results

### 3.1. Instinctual Survey

Of the *n* = 44 participants approached, only *n* = 15 instinctual surveys were completed. The HHACT was applied to the same case studies by *n* = 5 members of the Homeless Health Research Team. Moderate reliability (Kappa = 0.465) was observed between the fifteen raters for the instinctual ratings (Table 2). Good reliability (Kappa = 0.754) was reported between the five raters when the HHACT was used. There may be bias in comparing agreement between five and fifteen raters, as there may be less opportunity to disagree using a smaller sample size. To account for this, a sensitivity analysis was performed by random selection of five instinctual raters; this increased the reliability (Kappa = 0.609), but the agreement was lower among instinctual raters than among raters using the tool.

To compare the performance of instinctual and HHACT assessments, the lead author provided both an instinctual rating of the case studies and applied the HHACT. When the median of the fifteen instinctual raters was compared to the expert rating, there was moderate agreement (Kappa = 0.644), which increased when considering only five random ratings (Kappa = 0.800). Excellent agreement was observed between the median of the HHACT ratings and the expert (Kappa = 0.908). High agreement was also observed for total scores of the HHACT with an ICC of 0.839, with excellent reliability demonstrated between the expert and median score of raters 0.947 (Table 3).

The qualitative rationales provided for the ratings when applying the HHACT suggested ambiguity in the question “are you able to go for health care when you are not feeling well?”. Participants also highlighted the question regarding having a Medicare card and being eligible for Medicare and whether this may cause confusion. In addition, the term ‘squat’ was a suggested addition to the choice of locations to sleep, and ‘injection’ was suggested as an addition to the medication question. Adding an observation component to the questions related to disability was also suggested (Table 4).

### 3.2. Study 2

The VI-SPDAT and HHACT were administered to *n* = 20 participants residing in Tierney or Stanford House. The included participants captured a population that is reflective of the cohort of people experiencing homelessness in Australia. Of the 20 participants two were female and 18 were male, the average age was 50.6 years and four participants identified as Aboriginal. The predominant race was Australian (*n* = 10), Burmese (*n*−1), Aboriginal (*n* = 4), Irish (*n* = 1), Dutch (*n* = 1), Fiji Indian (*n* = 1), Korean (*n* = 1), English (*n* = 1) and three of the participants spoke a language other than English as their first language. The average number of diagnosed mental illnesses per participant was 2.45 and physical illness 3.85 and 9 participants reported problematic drug and alcohol use. The cumulative length of time spent homeless ranged from 5 months to 29 years and the average was 5.42 years. 12 of the participants reported experiencing trauma, 14 had a disability and three identified as LGBQIT+. The HHACT took between 5 min 6 s and 11 min 45 s to complete, the mean was 7 min 29 s (SD 118.03 s). All participants perceived the HHACT as easy to answer. Regarding length, *n* = 17 perceived it as just right, *n* = 2 as too short and *n* = 1 as too long. When asked whether the HHACT was able to determine health need and ability to access healthcare of a person experiencing homelessness, *n* = 19 agreed (one missing). Examples of the open-ended comments provided were, ‘*Just right, just enough information to keep attention span’*, ‘*They were in layman’s terms, generalized questions’, ‘Cause it was multiple choice it makes it easier to understand*.’ Four participants suggested some improvements to the HHACT, these included: (1) identifying whether a person has a partner, (2) focusing on immediate concerns is more important than historical issues, (3) use of a counselling framework to organize the questions, and (4) expansion of the section on medical issues. The HHACT was found to have convergent validity with moderate correlation observed between the VI- SPDAT and total HHACT, this was also the case for the risk’s subscale, lower correlation was observed for the other subscales.

### 3.3. Interview

The interview with the two colleagues administering the HHACT and Australian VI-SPDAT, lasted 34 min 25 s. The two participants reported feeling comfortable administering the HHACT, they perceived the questions as unintrusive and straightforward. The LGBQIT+ question was received awkwardly by those respondents that did not understand the associated terms, particularly the term ‘intersex’. In relation to the order of the questions, it was suggested that the Aboriginality question was moved higher up in the sequence of questions to signal its importance. Participants did not perceive that any of the questions were repetitive. It was recommended to include a question ‘are you receiving any supports currently?’. The length of the HHACT was considered appropriate, and respondents were able to maintain a high level of attention and engagement throughout. The participants suggested adding an observation component to the question related to disability, as some respondents denied having a disability and yet were in receipt of a disability support pension or registered with the national disability insurance scheme. Amendments to the wording of the questions ‘length of time experiencing homelessness’ and ‘ability to access healthcare’ and ‘do you have a Medicare card?’ were suggested. Proposed changes to the HHACT are summarized in Table 5.

## 4. Discussion

As part of the process of developing the HHACT these initial psychometric findings are promising in identifying that the HHACT has reliability and is feasible. Particularly important was testing the HHACT on people experiencing homelessness and exploring their perceptions of the wording of the questions and the length of time the questions took to answer. The time to complete the HHACT will be important when it is implemented in high tempo clinical settings such as emergency departments, therefore the fact that it takes approximately 7 min to complete is encouraging in terms of feasibility. The next step in testing is to implement the HHACT through a randomized controlled trial (RCT) at an emergency department in Melbourne. The trial has been registered with the Australian and New Zealand Clinical Trial Registry: https://www.anzctr.org.au/ACTRN12622001085763p.aspx (accessed on 26 November 2022).

In this next RCT study we will be able to undertake some further psychometrics including an exploratory factor analysis and qualitative feedback from healthcare professionals administering the HHACT. The RCT combines the HHACT with a Decision Assistance Guide, which will provide decision support to clinicians based on a person’s level of health-related vulnerability (low, moderate, high). The combination of the HHACT and the Decision Assistance Guide has been named the Homeless Health Response Bundle.

It is anticipated that implementation of the Homeless Health Response Bundle will facilitate access to healthcare for people experiencing homelessness by overcoming several of the challenges faced by emergency clinicians and people experiencing homelessness. Acknowledging that emergency departments are extremely busy, and the tempo of practice means that tackling the often-complex issues facing people experiencing homelessness can be highly challenging. People experiencing homelessness have reported that they leave the emergency department having had very few of their needs met [23].

In 2020 to 2021 the number of presentations to Australian Emergency Departments increased by 6.9% compared to 2019–2020 [29]. Given the increasing pressure on emergency departments the implementation of the Homeless Health Response Bundle aims to assist in early identification of people experiencing homelessness, thereby highlighting barriers to discharge, and exploring avenues to reduce readmission. A recent literature review of the prevalence of homelessness and the clinical reasons, and outcomes in relations to emergency department visits demonstrated that injury, mental health, and substance misuse-related concerns are the primary reasons for presentation to hospital [30]. Implementation of the Homeless Health Bundle will facilitate screening of homelessness and provide a standardized assessment of health-related vulnerability and will provide guidance to clinicians on how to respond. The primary outcome of implementation of the Homeless Health Response Bundle is to reduce reattendances to the emergency department within 28 days, by 20%. As far as the authors are aware there are no other interventions of this nature for people experiencing homelessness.

### Limitations

This study has some limitations. The sample sizes for both Study 1 and Study 2 were small and therefore provide initial indication as to the reliability and validity of the HHACT. The study was conducted between 2020–2021 during the initial and subsequent waves of the COVID-19 pandemic. We believe this impacted the response rate to the Study 1 instinctual survey, because workloads in the emergency department were particularly high, and resources such as staffing were variable. Low prevalence in the population for some conditions and behaviors meant that they were not observed in such a small sample, for example, none of the participants had daily functioning issues, which limited the ability to analyze the data with Cronbach’s alpha, this is not expected to be the case for larger sample sizes planned in the future. The qualitative findings from participants experiencing homelessness who received the HHACT were generally positive. Participants were provided with an incentive to undertake the study and therefore there is the possibility that selection bias occurred, and those that agreed to participate were more likely to have positive attitudes regarding the HHACT. Further, the participants were housed, albeit temporarily, during data collection and this may also have altered their responses to the HHACT.

## 5. Conclusions

Initial psychometrics suggest the use of the Homeless Health Access to Care Tool is feasible and reliable. Larger samples are required to report construct validity. Qualitative feedback from people experiencing homelessness suggest that the questions included in the Homeless Health Access to Care Tool are acceptable and appropriate. The time taken to complete the Tool was an average of 7 min, which suggests it would be feasible to use in high tempo clinical settings such as emergency departments and outreach settings. The next step in the development of the Homeless Health Access to Care Tool is to examine its construct validity by applying it to a larger sample size.

## Figures and Tables

**Figure 1 ijerph-19-15928-f001:**
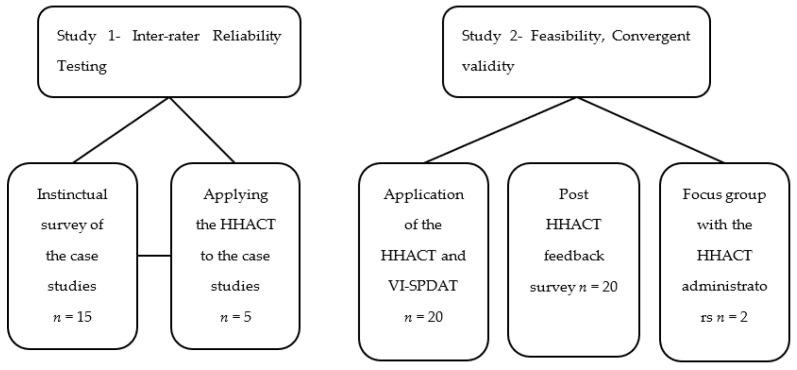
Study Flow Chart.

**Table 1 ijerph-19-15928-t001:** Original HHACT Survey [15] Re-use permitted under CC BY-NC., 2022, Jane Currie, et al.

Date of Survey: Completed by: Survey Location:	**Score**
What’s your name? Date of Birth: Age:	>55 yrs. = 1
What is your first language? Interpreter required?	Interpreter required = 1
What gender do you identify with? Male ☐ Female ☐ Other ☐ Prefer not to say☐ Don’t know ☐	Other, prefer not to say, don’t know = 1
What are your pro-nouns? She/Her ☐ He/Him☐ They/Them☐ Name only☐ Prefer to self-describe ☐	
Do you identify as LGBTIQ+ Prefer not to say ☐ Don’t know ☐ No ☐	LGBTIQ+, prefer not to say, don’t know Y = 1
Are you currently or could you be pregnant? Y ☐ N ☐	Y = 1
Where do you sleep most frequently and how long have you been staying there? The streets ☐ Train ☐ Car ☐ Crisis or emergency accommodation/ Shelter/ Refuge ☐ Staying with family or friends☐ Caravan☐ Hotel/motel☐ Hostel ☐ Boarding house ☐ Other☐: Specify	Streets, train or car = 1
What is the total time that you have experienced homelessness? Days ☐ Weeks ☐ Months ☐ Years ☐	> 6 months = 2
On a typical day what is the best way to contact you?	
Is there someone we can contact for you in an emergency? Y ☐ N ☐ Friend ☐ Relative ☐ Other: Name: Mobile number:	N = 1
Do you have a Medicare card? Y ☐ N ☐ Number:	N = 1
Do you identify as Aboriginal or Torres Strait Islander? Y ☐ N ☐	Y = 2
Are you or were any members of your community part of the Stolen Generation? Y ☐ N ☐	Y =2
Are you a refugee or seeking asylum? Y ☐ N ☐	Y = 1
Are you able to go for health care when you are not feeling well? Unable to go for, or avoids care? Y ☐ N ☐ Why?	N = 2
In the past six months, how many times have you: Received health care at an emergency department? Taken an ambulance to hospital? Been admitted to hospital? Spent time in prison or under police custody?	Yes to any on the list = 1 Yes to 2 or more on the list = 2
Are you currently able to take care of your daily needs like showering, changing clothes, using a toilet, getting food and something to drink? Y ☐ No ☐	N = 2
Observation: Does the client appear able to take care of daily needs? Y ☐ N ☐	N = 2
Do you get money from Centrelink ☐ A job☐ An inheritance☐ DVA ☐ Charity ☐ No income☐ Other: ☐ (specify)	No income = 1
Has a health professional told you that you have any medical conditions? A serious brain injury/head trauma ☐ Kidney disease/ dialysis ☐ Gastric disorders ☐ Liver disease/cirrhosis ☐ Heart disease ☐ High or low blood pressure ☐ Emphysema/COPD/asthma ☐ Diabetes ☐ Cancer ☐ Hepatitis C ☐ Epilepsy/seizures ☐ HIV/AIDs ☐ Heat stroke/exhaustion ☐ TB ☐ STI ☐ Physical injury ☐ Cellulitis ☐ Obesity ☐ Other: ☐ (specify)	Yes to 1 in the list = 1 Yes to 2 in the list = 2 Yes to 3 or more in the list
Are there any medications that you have been advised to have regularly? Y ☐ N ☐ What are they? Are you taking these as advised? Y ☐ N ☐ Can you tell us why not? Y ☐ N ☐ Can’t afford them ☐ Bad side effects ☐ They were stolen ☐ Unable to store them ☐ Forget to take them ☐ You don’t think you need them ☐ Other☐: (specify)	Y = 1 Not taking meds as advised = 1
Have you consumed alcohol Y ☐ N ☐ and/or drugs Y ☐ No ☐ almost every day or every day for the past month?	Y = 2
*Observation*: Does the person appear under the influence of drugs/alcohol now? Y ☐ N ☐ Does the person appear to be withdrawing from drugs/alcohol now? Y ☐ N ☐	Y to either = 2
Are you or has someone told you they are worried about your mental health? Y ☐ N ☐	Y = 1
Have you ever been diagnosed with a mental health condition? Y ☐ N ☐ Anxiety ☐ Depression ☐ PTSD ☐ Bipolar disorder ☐ Schizophrenia ☐ Psychosis ☐ Personality Disorder ☐ Cognitive impairment/dementia ☐ Other☐: (specify)	Yes to 1 in the list = 1 Yes to 2 in the list = 2 Yes to 3 in the list
Do you ever have thoughts of self- harm? Y ☐ N ☐	Y = 1
Do you ever have thoughts of suicide? Y ☐ N ☐ *Observation:*	Y = 1
Does the person demonstrate any signs and/or symptoms of a mental illness? Y ☐ N ☐	Y = 1
Have you ever been told that you have a disability? Y ☐ N ☐ Physical ☐ Intellectual ☐ Sensory ☐ Cognitive ☐ Psychosocial ☐ Receiving NDIS ☐ Receiving DSP ☐	Y = 2
*Note: Before asking this question, please consider if it is appropriate (safe) to do so*. Is there anyone that you feel unsafe with/threatened by or that causes you harm in any way? Y ☐ N ☐	Y = 1

**Table 2 ijerph-19-15928-t002:** Reliability of instinctual and HHACT ratings using Gwets Kappa.

Variable	*n*	Rater N	Raw Agreement	Weighted Agreement	Kappa (95% CI)	*p*-Value
Instinctual	18	15	51.5%	72.8%	0.465 (0.372, 0.558)	<0.001
HHACT	18	5	72.2%	86.1%	0.754 (0.542, 0.965)	<0.001
Expert vs. median Instinctual	18	2	66.7%	80.6%	0.644 (0.337, 0.951)	<0.001
Expert vs. median tool	18	2	88.9%	94.4%	0.908 (0.770, 1.000)	<0.001

**Table 3 ijerph-19-15928-t003:** Reliability HHACT scores using intraclass correlation.

Variable	*n*	Rater N	ICC (95% CI)	*p*-Value
HHACT Score	18	5	0.839 (0.713, 0.927)	<0.001
Expert vs. Median Tool Score	18	2	0.947 (0.813, 0.982)	<0.001

**Table 4 ijerph-19-15928-t004:** Convergent Validity.

	VI-SPDAT Med (IQR)	HHACT Med (IQR)	Spearman’s Correlation	95% CI Lower Upper		*p*-Value
Total scale	10.0 (8.5)	14.5 (6.3)	0.567	0.153	0.812	0.009
Demographics	0.0 (1.0)	1.0 (1.0)	0.295	−0.183	0.660	0.207
History of Housing and Homelessness	2.0 (0.0)	2.0 (1.0)	0.131	−0.343	0.552	0.582
Risks	2.0 (2.8)	3.0 (2.0)	0.560	0.143	0.809	0.010
Socialisation and Daily Functions	1.0 (1.8)	0.0 (1.0)	0.402	−0.063	0.724	0.079
Wellness	5.0 (5.8)	8.5 (2.8)	0.409	−0.055	0.728	0.073

**Table 5 ijerph-19-15928-t005:** Change made to HHACT Survey (highlighted in bold) [15] Re-use permitted under CC BY-NC., 2022, Jane Currie, et al.

Final Version	Final Scoring	Rationale for Changes
What gender do you identify with: Male ☐ Female ☐ **Non-binary ☐ I use a different term (please specify) ☐ Prefer not to answer ☐ Has this changed since birth? Yes ☐ No ☐** Do you identify as LGBTIQ+? Yes ☐ Prefer not to say☐ Don’t know ☐ No ☐	**Non-binary**, prefer not to say, **I use a different term = 1 Y = 1** LGBTIQ+, prefer not to say, don’t know Y = 1	Wording changed after the HHACT was administered to participants. Pro-nouns questions removed as didn’t change the degree of vulnerability and this populations would be captured with the other questions.
Are you currently or could you be pregnant? Yes ☐ N☐	**Y = 2**	Scoring increased -important indicator of vulnerability.
Where do you sleep most frequently and how long have you been staying there? The streets ☐ Train ☐ Car☐ Crisis or emergency accommodation/ Shelter/ Refuge☐ Staying with family or friends☐ Caravan☐ Hotel/motel☐ Hostel ☐ Boarding house ☐ Other☐ **Squat☐**: Specify What is the total time that you have experienced homelessness? Days☐ Weeks☐ Months☐ Years☐ On a typical day what is the best way to contact you?	Streets, train or car = 1 > 6 months = 2	Added squat as it is a common place for people experiencing homelessness to reside.
**Can you access Medicare? Yes** **☐** **No** **☐** **If no, why not?**	N = 1	Feedback about the difference between having a physical Medicare card and being eligible for Medicare.
Do you identify as Aboriginal or Torres Strait Islander: Yes ☐ No ☐? **Disclaimer: You don’t have to answer this if you don’t want to:** Are you or were any members of **your family or** community part of the Stolen Generation? Yes ☐ No ☐ Prefer not to answer ☐	Y = 2 Y **or prefer not to answer= 2**	This question was moved further up in the survey before the gender question. Added family based on feedback. Some participants found it to be a confronting question that they struggled to answer-so a disclaimer was included.
**Do you avoid or are you unable to go for care when you are not feeling well?** Unable to go for, or avoids care? Yes ☐ No ☐ Why?	Y = 2	In the feedback this was described as the most ambiguous question so was reverted to the original VI-SPDAT version. Scoring changed to from N = 2 to Y = 2.
In the past six months, how many times have you: Received health care at an emergency department? Taken by ambulance or **police** to hospital? Been admitted to hospital? Spent time in prison or under police custody?	Yes to any on the list = 1 Yes to 2 or more on the list = 2	Added in by ‘police’ based on feedback.
Are you currently able to take care of your daily needs like showering, changing clothes, using a toilet, getting food and something to drink? Yes ☐ No ☐ Observation: Does the client appear able to take care of daily needs? Yes ☐ No ☐ Do you get money from Centrelink ☐ A job☐ An inheritance☐ DVA **☐ DSP**☐ Charity ☐ No income☐ Other: ☐ (specify)?	No = 2 **Observation: No = 3 (score 3 if not already scored above)** No income = 1	Scoring changed to ensure that those who had insight into not being able to meet their daily needs didn’t receive a higher score. Added DSP
Are there any **injections** or tablets that you have been advised to have regularly? Yes ☐ No ☐ What are they? Are you taking these as advised? Yes ☐ No ☐Can you tell us why not? Yes ☐ No☐ Can’t afford them ☐ Bad side effects ☐They were stolen ☐ Unable to store them☐ Forget to take them ☐ You don’t think you need them ☐ Other☐: (specify)	Yes = 1 Not taking meds as advised = 1	Added in injections to ensure that medications such as depots and insulin were not missed- this was from the feedback.
Are you or has someone told you they are worried about your mental health? Yes ☐ No ☐Have you ever been diagnosed with a mental health condition? Anxiety ☐ Depression ☐ PTSD ☐ Bipolar Disorder ☐ Schizophrenia ☐ Psychosis ☐ Personality Disorder ☐ Cognitive impairment/dementia Other☐:(specify) **Do you have thoughts of self- harm**? Yes ☐ No ☐ **Do you have thoughts of suicide?** Yes ☐ No ☐ Observation: Does the person demonstrate any signs and/or symptoms of a mental illness? Yes ☐ No ☐	Yes = 1 Yes to 1 in the list =1 Yes to 2 in the list =2 Yes to 3 in the list = 3 Yes = 1 Yes = 1 Yes = 1	Changed the wording from “do you ever…” to “do you have” to create a timeframe for these thoughts and provide a clearer indication of vulnerability and risk.
Have you ever been told that you have a disability? Physical ☐ Intellectual ☐ Sensory ☐ Cognitive- **TBI ☐** Psychosocial ☐ Receiving NDIS☐ Observation: Does the person appear to have a disability? Yes ☐ No ☐Physical ☐ Intellectual ☐ Sensory ☐ Cognitive- TBI ☐ Psychosocial ☐	Yes to 1 in the list = 1 Yes to 2 or more in the list = 2 Yes = 2 (if not already scored above)	Added TBI and an observation question about disability to capture those who don’t have insight into/ express their disability.
Note: Before asking this question, please consider if it is appropriate (safe) to do so. Is there anyone that you feel unsafe with/threatened by or that causes you harm in any way? Yes ☐ No ☐	**Y = 2**	Scoring increased from 1–2 as this is a significant indicator of risk and vulnerability.

## Data Availability

Data are available from the corresponding author on reasonable request.

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
