# Peer review of "Multistage Psychometric Testing of the Homeless Health Access to Care Tool"

_ijerph, 2022, doi:10.3390/ijerph192315928_

Round 1

Reviewer 1 Report

This article presents the results of multistage testing and evaluation of a new Homeless Health Access to Care (HHACT) screening tool to be used by clinicians to prioritize and assess the needs of people who present as homeless in emergency departments (ED). The level of detail and development in this effort are significant and promising. I recommend publication of this important work. However, I have questions and/or suggestions for improvement/clarification.

Overall, the work was very thorough and represented a considerable detailed effort. However, the connection between the issues facing people experiencing homelessness and the clinicians serving them, and how the HHACT intends to address these issues and facilitate care provision by clinicians was not clearly articulated in the Introduction and Background. Only in the discussion did I see a mention of “decision support” as an objective of the entire endeavor of developing the HHACT. This should be more clearly articulated early in the paper.

While we can all agree that ED are not ideal places to provide adequate care to address the needs of individuals experiencing homelessness due to the rapid pace of needed care delivery, the demands of triage, etc., it was not clear how the HHACT (as opposed to the VI-SPDAT or instinctual evaluation) represents an improvement in ED triage and care to people experiencing homelessness. What does the HHACT do, or aim to do, that other approaches cannot do? How does the HHACT help clinicians provide more appropriate care? How will it provide decision support to clinicians? Does each set of questions produce a score that will help clinicians in the triage process?

Moreover, the presentation of the testing and the selection of test subjects at each stage was confusing and could have been more clearly presented. There were sections where I wondered how or why certain methodological decisions were made or carried out—which seems of greater importance in this paper:

Here are my suggestions and questions:

1.      How was the qualitative data from the instinctual rating obtained? I assume this was in written form, online. Did the clinicians provide enough detail for a qualitative analysis?

2.      It would have been helpful to provide a table to outline the 2 study phases and what was done in each. For example:

Reliability Testing- 44 clinicians invited, 15 clinicians completed instinctual survey to cases       

-        5 clinicians applied the HHACT

Feasibility Testing- 20 people experiencing homelessness residing in facilities completed the HHACT and VI-SPDAT; completed feedback survey afterwards (online?)

-2 researchers participated in a focus group about the survey administration experience

3.      In the feasibility testing, how was the team able to get 100% response to the feedback survey from the 20 people who sat for the administration of the HHACT? How was the survey administered, was it online or did they complete it immediately after the HHACT?

4.      In the table with the statistics, the term “Tool” is used to indicate the HHACT. Using different terms for the same thing adds confusion to an already complex process.

5.      Who was the “expert”? Why is the “expert” only introduced for the first time in the results and not in the methods if the analysis relies so heavily on this person’s assessment?

6.      Why wasn’t the larger population of 44 clinicians asked to qualitatively evaluate the HHACT, as opposed to the smaller group of 5 clinicians who comprise the Homeless Heath Research Team?

7.      A larger concern: it seems like the study had the Homeless Health Research Team of 5 clinicians apply and evaluate the HHACT; and 2 researchers participate in the focus group. It seems like the study just conducted the bulk of the research on the researchers’ impressions of using these tools, which was a missed opportunity to solicit feedback from the larger sample of 15 participating clinicians who completed the instinctual survey.

Author Response

Reviewer 1

1

How was the qualitative data from the instinctual rating obtained? I assume this was in written form, online. Did the clinicians provide enough detail for a qualitative analysis?

The qualitative data was obtained from free text comments in a survey housing the instinctual assessment of the case studies. After each of the case studies there was a question in the survey form which stated “please feel free to provide any comments in relation to the scoring of this case study.”  The participants provided rich feedback which was used in the qualitative analysis. The following sentences have been added to address this concern, Page 4 Line 148:

“After each case study the participants were asked to provide any free text comments in relation to the scoring of each case study. These free text comments provided a substantial amount of feedback regarding participants’ rationale underpinning their rating of each case study.”

 Page 4, line 176 The participants were also asked to provide free text feedback regarding their scoring of each case study.”

22  2

.      It would have been helpful to provide a table to outline the 2 study phases and what was done in each. For example:

Reliability Testing- 44 clinicians invited, 15 clinicians completed instinctual survey to cases       

-     5 clinicians applied the HHACT

Feasibility Testing- 20 people experiencing homelessness residing in facilities completed the HHACT and VI-SPDAT; completed feedback survey afterwards (online?)

-2 researchers participated in a focus group about the survey administration experience

We believe that Figure 1 sufficiently outlines the 2 study phases, as it includes the components of each phase and the number of participants included in each phase.

3

In the feasibility testing, how was the team able to get 100% response to the feedback survey from the 20 people who sat for the administration of the HHACT? How was the survey administered, was it online or did they complete it immediately after the HHACT? The team was able to get 100% response to the feedback survey as it was physically administered immediately after the HHACT. Amended as follows Page 7 line 228 “Immediately after the HHACT was administered, the Peer Support Worker administered a short feedback survey….”

44  4

.      In the table with the statistics, the term “Tool” is used to indicate the HHACT. Using different terms for the same thing adds confusion to an already complex process.

Who was the “expert”? Why is the “expert” only introduced for the first time in the results and not in the methods if the analysis relies so heavily on this person’s assessment?

Thank you, the word “Tool” has been changed to “HHACT” (Page 8 Table 2 and Table 3).

Thank you. Further explanation about why these participants are considered experts has been added into the Methodology section. Page 4, line 183 “members of the Homeless Health Research team who are clinicians and experts in the field were invited to apply the HHACT to each of the 18 case studies, via an online survey…”.

The following has been added to Page 5, line 186:

“The lead author, an expert in homeless health, undertook an assessment of each case study using their instinct and then by applying the HHACT. This expert assessment was undertaken to enable a comparison of the performance of instinctual and HHACT assessments.”

5

Why wasn’t the larger population of 44 clinicians asked to qualitatively evaluate the HHACT, as opposed to the smaller group of 5 clinicians who comprise the Homeless Heath Research Team?

The 5 members of the Homeless Health Research Team were asked to apply the HHACT to the case studies as they understood the tool and how it should be applied. The following has been added to increase clarity. Page 4, Line 185.

“These participants were purposively selected as they had an in depth understanding of the HHACT and consequently were best placed to apply it to the fictional case studies.”

6

A larger concern: it seems like the study had the Homeless Health Research Team of 5 clinicians apply and evaluate the HHACT; and 2 researchers participate in the focus group. It seems like the study just conducted the bulk of the research on the researchers’ impressions of using these tools, which was a missed opportunity to solicit feedback from the larger sample of 15 participating clinicians who completed the instinctual survey.

The clinicians in the Homeless Health Research Team were selected to apply the HHACT (Study 1) to the case studies as they had a good understanding of the HHACT. This understanding was not held by the other 15 participating clinicians. The researchers who administered the HHACT and VI-SPDAT to people experiencing homelessness (Study 2) were chosen specifically because they work in the homeless health service and have been trained in the use of the VI-SPDAT and have established trust in this homeless community.  

The next phase of testing of the HHACT involves application of the HHACT to n=100 participants in an emergency department in Melbourne via a randomised controlled trial. We will be seeking the views of these clinicians on their application of the HHACT.

Reviewer 2 Report

This paper describes feasibility and reliability testing of a tool to assess health vulnerabilities among people affected by homelessness. The authors note that existing assessments are focused on prioritizing housing despite there being a clear need to assess health vulnerabilities as well. The study is well designed, and the results are clearly described. This reviewer believes that the study makes a valuable contribution to the field.

Several minor revisions are recommended:

Page 1, line 39--the list of conditions associated with homelessness is not organized well. It is recommended to take out the phrase "illnesses related to rough sleeping" and replace it with "sleep deprivation"; it is also recommended to replace "mental ill health" with "declining mental health".

Page 1 line 42 this population "is", replace “are”

Page 2 line 82, delete "a"--Globally “a” 100 million  

Page 4 line 154, state the average years of experience or the range of years of experience, instead of claiming many years of experience for participants

Page 4 line 167 “Xcel”--I think this is Microsoft Excel

Page 7 line 230, only 15 out of 44 approached completed the assessment--if possible, state any known reasons for not completing the assessment.

page 12 line 341 rephrase to: that the “use of the HHACT” is feasible and reliable...

page 12 line 228 regarding limitations--describe the incentives in more detail--it is doubtful that the incentives changed opinion. However, it is likely that people that agreed to participate were more likely to have positive attitudes about the assessment (compared to those that did not participate) --so selection bias was likely present

The tool itself might benefit from an item about language proficiency--what is the first language, and can the person read and write in English. This could signal additional barriers to care and needs for support from interpreters, etc.

Author Response

Reviewer 2

7

Page 1, line 39--the list of conditions associated with homelessness is not organized well. It is recommended to take out the phrase "illnesses related to rough sleeping" and replace it with "sleep deprivation"; it is also recommended to replace "mental ill health" with "declining mental health".

We have not amended this sentence. Illnesses related to rough sleeping are more than sleep deprivation. We prefer to use the term mental ill health rather than declining mental health as the former identifies mental health as an illness.

8

Page 1 line 42 this population "is", replace “are”

Amended

9

Page 2 line 82, delete "a"--Globally “a” 100 million  

Amended

10

Page 4 line 154, state the average years of experience or the range of years of experience, instead of claiming many years of experience for participants

Amended as follows: Page 4, line 170

“…a group of clinicians, each with an average of greater than three years of experience…”

11

Page 4 line 167 “Xcel”--I think this is Microsoft Excel

Amended

12

Page 7 line 230, only 15 out of 44 approached completed the assessment--if possible, state any known reasons for not completing the assessment.

We have added an explanation to the Limitations as follows, Page 13, Line 387:

The study was conducted between 2020-2021 during the initial and subsequent waves of the COVID-19 pandemic. We believe this impacted the response rate to the Study 1 instinctual survey, because workloads in the emergency department were particularly high, and resources such as staffing were variable.”

13

page 12 line 341 rephrase to: that the “use of the HHACT” is feasible and reliable...

Amended

page 12 line 228 regarding limitations--describe the incentives in more detail--it is doubtful that the incentives changed opinion. However, it is likely that people that agreed to participate were more likely to have positive attitudes about the assessment (compared to those that did not participate) --so selection bias was likely present

Amended as follows, Page 13, line 396

“Participants were provided with an incentive to undertake the study and therefore there is the possibility that selection bias occurred, and those that agreed to participate were more likely to have positive attitudes regarding the HHACT.”

14

The tool itself might benefit from an item about language proficiency--what is the first language, and can the person read and write in English. This could signal additional barriers to care and needs for support from interpreters, etc.

Agreed. There is a question on the HHACT regarding whether an interpreter is required. Health literacy is a substantial issue among people experiencing or at risk of homelessness.

Reviewer 3 Report

Thank you for an interesting manuscript of an important topic. The manuscript has many strengths but to improve the manuscript I will focus of the parts that I think need to be developed.

1) A critical reflection and discussion about the use and consequences of the homeless health access to care tool is needed. In what way does the instrument improve ordinary care - can't the same or better results be obtained through person-centred care with a human-to -human dialogue? There are several studies with people in homeless requesting the personal encounter with continuity.

The definition and classification of vulnerability is meritorious but how can the needs described be met in practice, even if it is partly outside the purpose of this study, it affects the credibility of the instrument in a wider sense.

Aspects related to risks or burden with the instrument needs to be brought up e.g. related to stigmatization, whether the various questions are relevant to a person seeking emergency medical care - a major focus is on the social situation, possibility to meet basic needs for the person but the person is probably seeking care for some acute physical and/or mental problem - shouldn't that be the focus in the first place? How are patients that are homeless going to be detected? do you ask all patients if they are homeless? Even if there is a scientific rational for the instrument what will be the clinical explanation to patients why they need to fill in this kind of questions, when for instance seeking care experience acute heart failure or otitis etc ?

Is it relevant that several questions relate to identity and sexual orientation - this needs to be discussed and justified in relation to the health care needs of the patient?

2; The sample must be further described. How were the 20 residents selected - if the persons got 30 dollar it is quite well paid for completing the tool and giving some feedback - I suppose a lot of the residents would like to participate. The time to fill in the tool is short - that is of course good but could be questioned, especially when the sample is not described - among people in homelessness cognitive impairment is common, e.g. as a result of drug use and or violence/trauma, reading and language difficulties among people in homelessness is often described in other studies. Mental illness is another common problem for people in homelessness with problems that most probably will affect their capability to complete the tool. How the tool will or will not be used for people that are affected of alcohol- and or drugs are not explained - or how long it will take to complete the tool in those cases. All together is important to show that the test of the tool with people in homelessness is done with people representative for the people in homelessness - otherwise exclusion criteria need to be described

About the setting for the data collection with people in homelessness- the fact that the data collection for the study took place when the participant had roof over their head and perhaps also in a much more calm setting than it would be in a emergency unit, when the person most probably is stressed due to impaired health and perhaps also in combination of not knowing where to spend the night, has to be acknowledged and addressed as a limit of the study.

3, The data collection described as focus group needs to be reversed - 2 persons is not a group - and the participants are described as "colleagues" - colleagues to whom - what was the relationship between the researcher and the participants?

4; The response rate of 34 % among clinicians must be addressed and discussed - why was not more clinicians interested in participating and who reliably is the results based on this response rate?

Author Response

15

A critical reflection and discussion about the use and consequences of the homeless health access to care tool is needed. In what way does the instrument improve ordinary care - can't the same or better results be obtained through person-centred care with a human-to -human dialogue? There are several studies with people in homeless requesting the personal encounter with continuity.

We have addressed this as follows: Page 2, Line 69

“In future studies we intend to combine the HHACT with a decision assistance guide that will inform the onward care provided to people experiencing homelessness. By combining the HHACT and a decision assistance guide we intend that a comprehensive assessment is undertaken and that options for access to healthcare are maximised for each person experiencing or at risk of homelessness.”

16

The definition and classification of vulnerability is meritorious but how can the needs described be met in practice, even if it is partly outside the purpose of this study, it affects the credibility of the instrument in a wider sense.

We believe this is addressed in our response above.

17

Aspects related to risks or burden with the instrument needs to be brought up e.g. related to stigmatization, whether the various questions are relevant to a person seeking emergency medical care - a major focus is on the social situation, possibility to meet basic needs for the person but the person is probably seeking care for some acute physical and/or mental problem - shouldn't that be the focus in the first place? How are patients that are homeless going to be detected? do you ask all patients if they are homeless? Even if there is a scientific rational for the instrument what will be the clinical explanation to patients why they need to fill in this kind of questions, when for instance seeking care experience acute heart failure or otitis etc ?

These are valid questions but are beyond the scope of this paper. The focus of this paper is the initial psychometric testing of the HHACT rather than its implementation. The next phase of our study is an RCT to identify the impact of the HHACT and the Decision Assistance Guide (please see page 11 lines 305-309). Our next paper will discuss how the tool is used in practice and the explanation provided to patients. 

18

Is it relevant that several questions relate to identity and sexual orientation - this needs to be discussed and justified in relation to the health care needs of the patient?

The literature identifies those as LGBTIQ as higher risk of social marginalisation and less likely to access healthcare due to perceived social stigma.

19

The sample must be further described. How weDre the 20 residents selected - if the persons got 30 dollar it is quite well paid for completing the tool and giving some feedback - I suppose a lot of the residents would like to participate. The time to fill in the tool is short - that is of course good but could be questioned, especially when the sample is not described - among people in homelessness cognitive impairment is common, e.g. as a result of drug use and or violence/trauma, reading and language difficulties among people in homelessness is often described in other studies. Mental illness is another common problem for people in homelessness with problems that most probably will affect their capability to complete the tool. How the tool will or will not be used for people that are affected of alcohol- and or drugs are not explained - or how long it will take to complete the tool in those cases. All together is important to show that the test of the tool with people in homelessness is done with people representative for the people in homelessness - otherwise exclusion criteria need to be described

Thank you. We have added some participant demographics to Page 9, Line 297 as follows: “The included participants captured a population that is reflective of the cohort of people experiencing homelessness in Australia. Of the 20 participants 2 were female and 18 were male, the average age was 50.6 years and 4 participants identified as Aboriginal. The predominant race was Australian (n=10), Burmese (n-1), Aboriginal (n=4), Irish (n=1), Dutch (n=1), Fiji Indian (n=1), Korean (n=1), English and 3 of the participants spoke a language other than English as their first language. The average number of diagnosed mental illnesses per participant was 2.45, physical illnesses 3.85 and 9 participants reported a history of problematic drug and/or alcohol use. The cumulative length of time spent homeless ranged from 5 months to 29 years and the average was 5.42 years. 12 of the participants reported experiencing trauma, 14 had a disability and 3 identified as LGBQIT+.”

20

About the setting for the data collection with people in homelessness- the fact that the data collection for the study took place when the participant had roof over their head and perhaps also in a much more calm setting than it would be in a emergency unit, when the person most probably is stressed due to impaired health and perhaps also in combination of not knowing where to spend the night, has to be acknowledged and addressed as a limit of the study.

We have added the following Page 13, line 398.

“Further, the participants were housed, albeit temporarily during data collection and this may also have altered their responses to the HHACT. “

21

The data collection described as focus group needs to be reversed - 2 persons is not a group - and the participants are described as "colleagues" - colleagues to whom - what was the relationship between the researcher and the participants?

The term focus group has been changed to interview. Page 10, line 324. 

22

The response rate of 34 % among clinicians must be addressed and discussed - why was not more clinicians interested in participating and who reliably is the results based on this response rate?

We have added an explanation to the Limitations as follows, Page 13, Line 387: “The study was conducted between 2020-2021 during the initial and subsequent waves of the COVID-19 pandemic. We believe this impacted the response rate to the Study 1 instinctual survey, because workloads in the emergency department were particularly high, and resources such as staffing were variable.”

Round 2

Reviewer 3 Report

Thank you for your responses - I found the manuscript improved and hope you in comming studies have the possibility to address the comments I made that could not be captured in this manuscript. Wish you good luck with the development and implication of the tool.